# Apoplastic and Symplasmic Markers of Somatic Embryogenesis

**DOI:** 10.3390/plants12101951

**Published:** 2023-05-11

**Authors:** Ewa Kurczynska, Kamila Godel-Jędrychowska

**Affiliations:** Institute of Biology, Biotechnology and Environmental Protection, Faculty of Natural Sciences, University of Silesia, ul. Bankowa 9, 40-007 Katowice, Poland

**Keywords:** arabinogalactan proteins, cell wall, pectins, plasmodesmata, lipid transfer proteins, symplasmic communication, somatic embryogenesis

## Abstract

Somatic embryogenesis (SE) is a process that scientists have been trying to understand for many years because, on the one hand, it is a manifestation of the totipotency of plant cells, so it enables the study of the mechanisms regulating this process, and, on the other hand, it is an important method of plant propagation. Using SE in basic research and in practice is invaluable. This article describes the latest, but also historical, information on changes in the chemical composition of the cell wall during the transition of cells from the somatic to embryogenic state, and the importance of symplasmic communication during SE. Among wall chemical components, different pectic, AGP, extensin epitopes, and lipid transfer proteins have been discussed as potential apoplastic markers of explant cells during the acquisition of embryogenic competence. The role of symplasmic communication/isolation during SE has also been discussed, paying particular attention to the formation of symplasmic domains within and between cells that carry out different developmental processes. Information about the number and functionality of plasmodesmata (PD) and callose deposition as the main player in symplasmic isolation has also been presented.

## 1. Introduction

### 1.1. Somatic Embryogenesis

The shortest definition of somatic embryogenesis (SE) is the formation of somatic embryos from somatic cells without fertilization [1,2,3]. During SE, somatic cells acquire an embryonic character due to the dedifferentiation of non-zygotic cells, the activation of cell division, and the reprogramming of the cell’s fate (for review, see [4,5,6]). The SE can be divided into several stages: induction, expression, development, maturation, germination, and plant conversion [5]. Many factors regulate SE. Among them are plant growth regulators [7], with auxin as a prominent player and a significant trigger for the acquisition of embryogenic competence within explant cells (e.g., [8,9,10,11]). Genetic and epigenetic changes associated with SE are well documented [11,12,13].

Moreover, changes in cell fate during SE are believed to be a reaction to stress conditions [11]. During such stress, homeostasis, e.g., of the balance between the capacity of and demand for proteins, up- and downregulated genes, and epigenetic changes, is altered, resulting in changes in the developmental process and the entry pathway of SE. It is also postulated that cell wall composition alteration and cell-to-cell signaling regulate SE [14]. 

### 1.2. Importance of Information Exchange

In the case of a multicellular organism, the exchange of information between cells is an important factor regulating its development, i.e., the formation of appropriate phenotypes of cells that make up this organism. The information exchange between cells occurs through various mechanisms and at different levels of the organization, including symplast and apoplast. The apoplast is a continuous system of cell walls. The symplast is a continuous system of cell protoplasm and cell membrane connected by plasmodesmata (PD). Cell-to-cell communication is an essential parameter coordinating the development of a multicellular organism, both for animals and plants. Creating spatial patterns and specifying the developmental program of individual cells requires an exchange of information, with this simultaneously being positional information, which is particularly important in plants. In the case of plant organisms, the positional information, i.e., the position of the cell in the organ, determines which path of development a given cell will take, and which set of chemical and physical factors resulting from its vicinity to other cells will affect it. This paper describes the involvement of the cell wall and symplasmic communication as elements of cell-to-cell information exchange during SE.

## 2. Cell Wall Composition as a Marker of Changes in Cell Fate during SE

The cell wall is a highly dynamic structure of plant cells. It undergoes remodeling under the influence of different signals, including changes during the acquisition of the embryogenic competence of cells during SE. The cell wall consists of cellulose microfibrils embedded in a hydrated matrix of polysaccharides such as hemicelluloses, pectins, and structural proteins [15,16,17,18]. In recent years, new techniques (transcriptome and proteome analysis) have confirmed changes in gene expression associated with wall remodeling during SE [19,20]. For example, it is postulated that the initiation of SE is correlated with an increased level of reactive oxygen species (ROS) and other stress-related factors, as well as genes encoding cell wall remodeling factors, such as expansin, extensin, pectinesterase, and glucanase, which are upregulated, and can play a relevant role during the acquisition of SE [6]. Moreover, some wall components have signaling properties, e.g., arabinogalactan proteins (AGPs) [19] and pectin oligosaccharide fragments [21]. An analysis of the SE of *Citrus sinensis* showed transcriptome changes, including reverse glycosylating protein (RGP-1), which is involved in polysaccharide metabolism in the early phases of SE [22]. 

Nowadays, diverse antibodies are used to precisely determine the cellular and subcellular distribution of cell wall components. The results from this type of research will be presented below.

### 2.1. Pectins

Pectins are a group of widely distributed plant cell wall polysaccharides containing galacturonic acid linked at positions 1 and 4. A detailed description of the structure of pectins and their transformations in the cell wall during developmental processes is well-known and described in many publications [17,23,24]. Some pectic epitopes studied so far indicate that they can be considered as markers of cell transition from somatic to embryogenic states (Table 1).

A pectic epitope recognized by the LM6 antibody (arabinan side chains) was detected in the embryogenic calli of *Brachypodium distachyon* [25]. Analyses of *Daucus carota* SE revealed that LM6 pectic epitope was a positive marker of cell reprogramming to the meristematic/pluripotent state, but the LM5 pectic epitope (galactan side chain; Figure 1—an example of localization of this epitope in explant) was postulated as a negative marker [26]. Similar results were described for *Trifolium nigrescens* explants during SE, where the LM5 galactan epitope was not detected in embryogenic protrusions, but LM6-reactive pectins were abundantly present in the cells of embryogenic swellings [27]. LM6 antibody was also detected during *Quercus suber* SE. The lowest labeling intensity was observed in the proembryogenic mass compared to the increase in signal intensity observed in somatic embryos [28]. On the other hand, during *Arabidopsis thaliana* SE, the LM5 epitope was detected within the embryogenic area of the explant [29]. The presence of arabinan and galactan pectin side chains was studied during the microspore embryogenesis of *Brassica napus.* During this process, diverse amounts of the LM5 and LM6 epitopes were described [30]. Both arabinan and galactan pectin side chains undergo modification under diverse factors [31]. The function of pectin arabinan and galactan side chains is not fully understood. It is postulated that arabinan facilitates wall rehydration and that arabinans function as pectic plasticizers, promoting cell wall flexibility [32]. It is proposed that in the walls of cells undergoing intensive divisions, the presence of arabinans side chains can serve as a marker of cell division, and the presence of galactan pectin side chains is a marker of cell elongation. It can be speculated that both arabinan and galactan may be markers for SE since cell division and growth occur during this process. The observed differences in the data described above may result from the analyses carried out at different time points of SE induction. Further research is needed to identify wall markers of cell transition from a somatic to an embryogenic state.

It is postulated that different pectin esterification levels correlate with the acquisition of embryogenic competence in explant cells [33,34,35]. An analysis of pectins with different levels of esterification indicated that highly esterified pectins were predominant in the walls of cells implementing the SE process. For example, during *Quercus alba* SE, it was determined that esterified pectins are characteristic for embryogenic cells [36]. In Arabidopsis explants, embryogenic cells are also characterized by the presence of highly esterified pectins [29]. Similar results were described for the embryogenic calli of *B. distachyon* [25]. During the SE of *Q. suber*, a spatiotemporal analysis of the presence of pectic epitopes recognized esterified (JIM7, LM20) and de-esterified pectins (JIM5, LM19) in the walls of embryogenic cells; this showed that at the beginning of SE, the proembryogenic masses were characterized by the abundant presence of esterified pectins, and low amounts of de-esterified pectins [28]. On the other hand, during *Musa* spp. [37] and Cichorium [33] SE, the embryogenic cells were rich in the JIM5 (low esterified pectins) epitope, but the JIM7 epitope (high esterified pectins) was less frequently represented. Moreover, the LM19 and LM20 (esterified and nonesterified pectins, respectively) antibodies characterized the cells in the embryogenic cultures of Brahypodium [25,38].

These small amounts of literature data (small because studies of this kind are carried out to a small extent) indicate some discrepancies, and may suggest that markers of changes in the direction of cell differentiation during SE cannot be identified using pectic epitopes for this purpose. This variation in results may be due to the analysis of different species, different culture conditions, and different sampling times during the culture. It seems that if we do not have, for example, the location of the *WOX* gene, it is difficult to precisely answer which of these epitopes is a marker of cell transition to the embryogenic state.

### 2.2. Extensins

Extensins belong to hydroxyproline-rich glycoproteins (HRGPs) which are engaged in different developmental processes, including SE ([25,37,39]; Table 1). The JIM11 and JIM20 antibodies (recognize specific arabinosylation motifs of HRGPs) were used to determine extensin localization in explant cells during SE [40]. In the case of *Musa* spp., JIM11 and JIM20 HRGPs epitopes were found in the early stages of SE in walls of embryogenic cell [40]. An analysis of extensin during *Dactylis glomerata* SE showed that the JIM12 antibody marks the cells in embryogenic calli [41]. Studies on Phalaenopsis culture revealed that the JIM11 and JIM20 HRGPs could be used as positive markers for cells with embryogenic competence in different orchid callus cultures [42]. For Brachypodium culture, the presence of an extensin that is recognized by the JIM11 antibody was detected, but extensin recognized by the JIM12 antibody was not present in embryogenic cells [25]. In contrast, studies on *Actinidia arguta* in vitro culture showed that embryogenic and non-embryogenic calli were rich in extensin epitopes recognized by the JIM11, JIM12, and JIM20 antibodies [43]. Because there are little literature data on the role of extensins in the regulation of SE, it is impossible to present a single scenario of their participation in changing the direction of cell differentiation during SE, and thus further research is necessary.

### 2.3. Arabinogalactan Proteins (AGPs)

The involvement of AGPs in SE, both during the early stages and somatic embryo development, has been widely described (excellent reviews were provided by [44,45]). Therefore, this review will only supplement the information already available, with particular emphasis on the immunolocalization of individual AGP epitopes in explant cells, changing the direction of differentiation to an embryogenic state, and supplemented with the current results on gene expression. 

The most frequently analyzed AGP epitopes in their role in SE were JIM8, JIM4, LM2, JIM13, and JIM16 (Table 1). The AGPs recognized by LM2 antibodies were detected at all stages of *Q. suber* SE. With the progression of SE, an increase in signal intensity was observed in somatic embryos in comparison to the lowest labeling intensity in the proembryogenic mass [28]. Studies of *Areca catechu* SE indicate that AGPs and cell wall lignification may be key steps for SE [46]. An analysis of the embryogenic callus of *B. distahyon* revealed that AGP epitopes, which were recognized by the JIM16 and LM2 antibodies, were positive markers of embryogenic cells [25]. For the *Fagopyrum tataricum* embryogenic callus, it was demonstrated that the AGPs epitope recognized by the LM2 antibody was a positive marker of the embryogenic cells [47]. Studies during the SE of Arabidopsis reported that the JIM16 AGP epitope appeared to be a marker of embryogenic cells [29]. The same studies showed that JIM4, JIM8, and LM2 antibodies were not wall components of embryogenic cells and thus can be considered as negative markers of cells differentiating to an embryogenic state. An analysis of the distribution of AGPs in explant cell walls during *D. carota* SE revealed that β-linked GlcA (LM2) was a positive marker of cell reprogramming to the meristematic/pluripotent state, and the JIM8 and JIM13 AGPs epitopes were negative markers of embryogenic cells [26]. It is postulated that the JIM8 antibody can be considered as a negative marker of embryogenic cells. Studies on carrot SE showed that this epitope does not coincide with the ability of individual suspension cells to form embryos [48], and that only cells without this epitope develop into somatic embryos [49]. During the microspore embryogenesis of *Brassica napus*, it was shown that JIM13 was a marker of totipotent cells [30,50].

Undoubtedly, the AGPs are involved in the SE process. However, much more research needs to be conducted in order to obtain a complete picture of AGPs role in redirecting cell differentiation during SE. Together with other components of cell walls, they contribute to wall remodeling and may be involved in changes in the physical properties of the wall. AGPs are tethered to the plasma membrane by the lipid glycosylphosphatidyl inositol anchor (GPI), and, after cleavage, they can be distinguished from the plasma membrane; it is postulated that then AGP is a signaling molecule [51]. Moreover, AGPs are proposed to play a role as pectic plasticizers, which can participate in the loosening of the pectic network [52].

**Table 1 plants-12-01951-t001:** Summary of the epitopes investigated in SE.

Epitope	Characteristic	Species	Cell Type Marker
Pectin			
LM6	Linear pentasaccharide in (1–5)-α-l-arabinans (RG I side chain	*Brachypodium distachyon*	embryogenic calli [25]
		*Daucus carota*	meristematic/pluripotent cells [26]
		*35S:BBM Arabidopsis thaliana*	embryogenic cells
		*Trifolium nigrescens*	embryogenic swellings [27]
		*Brassica napus*	embryogenic microspore, pollen-like structure [30]
		*Quercus suber*	proembryogenic masses, somatic embryos [28]
LM5	Linear tetrasaccharide in (1–4)-β-D-galactans (RG I side chain)	*Arabidopsis thaliana*	cells in embryogenic protrusions [29]
		*Brassica napus*	embryogenic microspore, pollen-like structure [30]
		*Trifolium nigrescens*	nonembryonic cells [27]
		*35S:BBM Arabidopsis thaliana*	explant cells
		*Daucus carota*	nonembryonic cells [26]
JIM7	Highly methyl-esterified HG	* Quercus alba *	embryogenic cells [36]
		*Brachypodium distachyon*	embryogenic calli [25]
		*Arabidopsis thaliana*	embryogenic cells [29]
		*Quercus suber*	proembryogenic masses [28]
		*Musa* spp.	nonembryonic cells [37]
		*Cichorium* hybrid	nonembryonic cells [33]
JIM5	Low methyl-esterified HG	*Musa* spp.	embryogenic cells [37]
		*Cichorium* hybrid	embryogenic cells [33]
		*Quercus suber*	advanced stages of somatic embryogenesis [28]
LM19	Homogalacturonan (HG) domain in pectic polysaccharides, which recognizes a range of HGs with preferential binding to unesterified HGs	*Brachypodium distachyon*	embryogenic cells [25,38]
		*Quercus suber*	advanced stages of somatic embryogenesis [28]
LM20	HG domain in pectic polysaccharides, which requires methyl esters for recognition of HG and binding to esterified HGs	*Brachypodium distachyon*	embryogenic cells [25,38]
		*Quercus suber*	proembryogenic masses [28]
Extensin			
JIM11	Extensin/HRGP glycoprotein	*Musa* spp.	embryogenic cells [40]
		Phalaenopsis	embryogenic cells [42]
		*Brachypodium distachyon*	embryogenic calli [25]
		*Actinidia arguta*	embryogenic and nonembryonic cells [43]
JIM20	Extensin/HRGP glycoprotein	*Musa* spp.	embryogenic cells and nonembrygenc cells [40]
		Phalaenopsis	embryogenic cells [42]
		*Brachypodium distachyon*	embryogenic calli [25]
		*Actinidia arguta*	embryogenic and nonembryonic cells [43]
JIM12	Extensin/HRGP glycoprotein	*Dactylis glomerata*	embryogenic calli [41]
		*Actinidia arguta*	embryogenic and nonembryonic cells [43]
		*Brachypodium distachyon*	nonembryonic cells [25]
AGP			
JIM8	Arabinogalactan	*Arabidopsis thaliana*	explant cells [29]
		*Daucus carrota*	nonembryonic cells [26]
JIM4	Arabinogalactan glycoprotein, βGlcA-(1,3)-αGalA-(1,2)-Rha	*Arabidopsis thaliana*	explant cells [29]
LM2	Arabinogalactan protein, carbohydrate epitope containing β-linked GlcA	*Quercus suber*	proembryogenic masses, somatic embryos [28]
		*Brachypodium distachyon*	embryogenic calli [25]
		*Fagopyrum tataricum*	embryogenic cells [47]
		*Arabidopsis thaliana*	explant cells [29]
		*Daucus carrota*	meristematic/pluripotent cells [26]
JIM13	Arabinogalactan/Arabinogalactan protein, carbohydrate epitope (β)GlcA1->3(α)GalA1->2Rha	*Daucus carrota*	nonembryonic cells [26]
		*Brassica napus*	totipotent cells [30,49]
JIM16	Arabinogalactan/Arabinogalactan protein	*Brachypodium distachyon*	embryogenic calli [25]
		*Arabidopsis thaliana*	embryogenic cells [29]

### 2.4. Lipid Transfer Proteins (LTPs)

One of the wall components neglected in research on wall remodeling during SE is the LTPs. It is postulated that LTPs are involved in cutin transport and polymerization. It was shown that in *Cichorium hybrid*, *Medicago sativa*, *Picea abies*, *D. carota,* and *A. thaliana* [52,53,54,55,56], LTPs were an early marker of the induction of SE. An analysis of the gene expression during *Triticum aestivum* SE indicates that LTPs were also involved during the acquisition of embryogenic character [57]. A transcriptomic analysis of *Gossypium hirsutum* revealed that embryogenic calli were characterized by the up-regulation of many genes, including *LTPs,* compared to non-embryogenic calli [58,59]. This analysis indicates that LTPs drive the totipotency of somatic cells, at least in cotton. 

### 2.5. Summary of Wall Components in SE and Future Prospects

SE can start in many different ways, and likely there is more than one molecular mechanism involved in these changes [60]. Accumulating data indicate that changes in cell fate are “visible” in the molecular structure of the cell wall. It can be postulated that it is the universal response of cells that accompanies the changes in their fate. However, it must be taken into account that not necessarily the same epitopes of pectins, AGP, and extensins will be markers of cell transition from the somatic to the embryogenic state for each species and tissue undergoing SE. It is well known that the explants used for SE induction can be various plant fragments, from the zygotic embryo to the root and shoot parts, which are undoubtedly characterized by different constitutive wall chemical components. That is why, in these analyses, special attention should be paid not so much to the current chemical composition of the wall but to the changes that occur in this composition during the transition from somatic to embryogenic cell state.

Moreover, the exposed differences in cell wall markers described so far may be due to the fact that the research was carried out on monocot and dicot plants, for which we know have differing wall chemical compositions [40]. For example, an analysis of the monosaccharide composition of the primary cell walls of 22 angiosperms showed differences in pectin content in dicotyledon and monocotyledon species [61]. Other studies showed that pectins from different monocot and dicot species differ in regard to their macromolecular structures. It was proven that in monocot, such as *Ananas comusus* and *Musa AAA* cv. Poyo, cell walls contain pectins richer in rhamnogalacturonan I but less rich in homogalacturonan building blocks compared to the dicot cell wall pectins [62]. The differences found, therefore, result from the systematic position of the species studied.

Further studies are needed, not only at the gene expression level but combined with immunolocalization in cells during SE. The need to supplement molecular analyses with immunochemical tests results from the fact that, for example, biochemical or gene expression methods give us general information about the chemical composition of cell walls and changes in gene expression during SE. However, collecting large amounts of material, generally including a diversity of cell types, is necessary for this type of research. It is therefore impossible to make these analyses only for embryogenic or meristematic cells, which are always in the vicinity of cells that do not participate in SE. Thus, the study of each cell type is missing. Therefore, an analysis using immunohistochemical methods is necessary in order to understand the molecular differences between cell walls and domains of local organization within cell walls [63]. It is not questioned whether immunohistochemical methods allow us to understand the organization and architecture of the wall of an individual cell and determine differences between cell walls associated with the development of cell types [63].

## 3. Symplasmic Isolation as a Mechanism Controlling Somatic Embryogenesis

One of the ways of exchanging information between cells is PD. Determining which cells and tissues communicate via PD reveals when and where differentiation signals are transmitted. The exchange of information through PD is called symplasmic communication. It is known that proteins (including transcription factors, TF) and different classes of ribonucleic acids (RNA) can pass through PD. Of the 2000 TF identified in Arabidopsis, several hundred are believed to be able to translocate through PD [64]. Other performed studies analyzed 76 TF labeled with mCherry, and 22 of them were translocated through PD [65]. So far, TF and/or RNAs have not been shown to move through PD during SE. However, the demonstration that other proteins, including GFP [66,67], are translocated during SE indicates that the cell-to-cell movement of TF or different classes of RNA may also occur during this process. The movements through PD of mobile TF and different classes of RNAs are well documented. For example, the movement of SHORT-ROOT (SHR) TF is necessary for endodermis specification [68], and CAPRICE (CPC) is vital during the specification of trichoblasts and antrichoblasts [69]. The movement of KNOTTED1 (KN1) TF through PD is polarized because it proceeds from the inner layers of the shoot apical meristem to the L1 layer, which indicates that it participated in maintaining the shoot stem cells [70,71]. The obtained results on different classes of RNAs, e.g., miR165a and miR166b which are involved in the development of vascular tissue in *A. thaliana* root, also showed their movement through PD [72,73]. All these data indicate that the impact of symplasmic communication on the movement of signaling molecules in the regulation of cell differentiation is not questioned. Unfortunately, as pointed out above, no data show which TF and/or RNAs are translocated through PD during SE, but it seems to be only a matter of time until we find out which signaling molecules move through PD. 

Since the molecules mentioned above have a significant impact on cell differentiation, it can be assumed that the regulation of their movement by changing the PD SEL (functional size exclusion limit) [74,75] may affect the regulation of cell differentiation. Changes in symplasmic communication by changing the number of PD and their capacity to translocate signals indicate a precise regulation of this process. This means that PD and symplasmic communication are important in cell differentiation and plant morphogenesis. Classical studies have shown that changes in the gene expression program are correlated with changes in symplasmic communication, expressed by changes in PD number or their permeability. Examples of such a relationship are as follows: (1) *Chara vulgaris* L.—loss of plasmodesmal connections determines the proper course of spermatogenesis [76]; (2) *Onoclea sensibilis* fern prothallus—blocking communication via PD results in cell totipotency, and, consequently, the production of a new prothallus by these cells [77].

Due to the existence of PD, it could be assumed that the plant body is a single symplasm, i.e., all cells throughout the plant’s life are connected through functional PD. However, it turns out that the plant body is divided into independent functional subunits called symplasmic domains. The symplasmic domain is an area that may consist of a single cell, many cells, or even an entire organ. Within the symplasmic domain, the cells are connected by functional PD but not connected by functional PD to surrounding tissues. Symplasmic domains can be permanent (e.g., stomata) and temporal, enabling cell, tissue, and proper organ differentiation [67,73,78,79]. Much literature data indicate that symplasmic domains are correlated with developmental processes. Studies conducted so far suggest that the changes in the direction of cell differentiation are accompanied by changes in symplasmic communication, mainly based on symplasmic isolation. It was proven that symplasmic isolation often precedes, or coincides with, initiating cell fate changes, suggesting it is required for cell differentiation [80].

Symplasmic domains were determined for embryogenesis [66,81,82,83,84], androgenesis [85], and somatic embryogenesis [67,86,87,88,89,90]. 

The most reliable studies of symplasmic communication use the analysis of the movement of fluorochromes, dextrans, or proteins within the explant, and such tests are increasingly undertaken. Unfortunately, information on symplasmic communication changes during embryogenesis is scarce.

The correlation between symplasmic domains and zygotic embryo development has been described for Arabidopsis and Sedum [66,82,83,88,89]. Studies on Arabidopsis showed that the zygotic embryo is a single symplasmic domain during all stages of development, but only for low molecular weight molecules (up to 0,5 kDa). With progress in embryo development, the symplasmic domain correlated (determined based on GFP movement) with organ and tissue differentiation [66,82,83]. It was also shown that the size of the GFP reporter molecules that can pass through PD was lowered during embryo development [66]. Comparisons of GFP (construct of varying sizes) movement in zygotic and somatic embryos of Arabidopsis showed that the symplasmic domains form similarly in both types of embryos. However, there are some qualitative differences such as the timing of establishing domain boundaries and the sizes of molecules that can move between symplasmic domains [88,91].

Studies on symplasmic communication during SE were performed using the example of direct somatic embryogenesis in the tree fern *Cyathea delgadii* [87]. An analysis using low-molecular-weight fluorochromes showed that the restriction in symplasmic communication preceded the induction of SE. During somatic embryo development, symplasmic domains appeared and they corresponded to the four segments of the somatic embryo [87]. Symplasmic communication during SE was also performed using various fluorescent tracers with different molecular weights, dextrans, and cytoplasm-soluble GFP expressed under the control of an STM promoter in the example of Arabidopsis. These studies showed a correlation between symplasmic isolation and cell fate changes during somatic embryo induction. At the beginning of the SE, the symplasmic communication of explant cells was intensive but decreased in regions undergoing different developmental fates, and embryogenic cells were isolated symplasmically from non-embryogenic ones [67]. In these studies, an analysis of the expression of the *WOX2* gene was performed. It showed that it correlated with areas of explant cells isolated symplasmically, which are a source of somatic embryos. Moreover, it was documented that callose biosynthesis is required to initiate SE and is correlated with decreased *DR5* auxin response in embryogenic cells [67].

### 3.1. Callose

As mentioned above, callose is involved in cell differentiation, and thus the cell fate changes. Indeed, callose is postulated to be the main factor controlling the formation of symplasmic domains (Figure 2). Callose (β-1,3-glucan polysaccharide) accumulates in PD in response to developmental and environmental signals (biotic and abiotic factors) (for review, see [92]). Callose in PD constricts the pore, inhibiting intercellular trafficking. Callose is often deposited in PD during different developmental processes as a subset of cells sequestered from their neighbors in order to allow the realization of a specialized developmental program. The amount of callose deposited depends on many factors, mainly on the opposing effect of the activity of two enzymes (callose synthase and callose-degrading β-1,3-glucanases) but also on many proteins related to PD and gene expression [93]. Callose deposition in PD is associated with the organelle–nucleus–plasmodesmata signaling (ONPS) pathway, where the amount of callose affects symplasmic communication by changing the level of expression of different genes [94]. A detailed description of callose turnover in PD is provided by Amsbury et al. [95]. Emerging data indicate that callose is the main “player” in regulating PD permeability. Despite the large number of publications describing the participation of callose in PD regulation, there are, unfortunately, not many studies that examine its role in somatic embryogenesis, specifically for callose deposition and the determination of the embryogenic competence of explant cells. The first information about the relationship between the callose and somatic embryogenesis came from studies in Cichorium and *Camellia japonica*, which showed that the callose deposition precedes the acquisition of embryogenic features of explant cells [96,97,98,99,100,101]. From that moment, we have seen many results that confirm the regulation of symplasmic communication by callose, including results indicating its significant SE regulation. An analysis of the callose deposition during the SE of Arabidopsis showed that cells that changed the direction of differentiation were isolated by callose [102].

Further research on Arabidopsis SE using the *35S:BBM WOX2:NLS-YFP* transgenic line showed that callose deposition in PD preceded *WOX2* gene expression in future sites of somatic embryo development [67]. Moreover, blocking callose synthesis suppressed somatic embryo development. The deposition of callose after plasmolyzing treatment was shown for *Eleutherococcus senticosus* SE [103] and *Syagrus oleracea* [104], indicating that the physiological and physical isolation of explant cells stimulates the reprogramming of somatic cells of the explant to embryogenically competent cells, which are a source of somatic embryos. In the case of *Araucaria angustifolia* calli, the involvement of callose as an element of SE regulation is also postulated [105]. Together these data indicate that callose deposition at PD is required for symplasmic isolation and the establishment of cell totipotency during SE. 

### 3.2. PD Number and Symplasmic Communication

The movement of signals through PD is regulated not only by the regulation of PD SEL but also by the number of PD between cells implementing the same or different developmental programs. Analyses performed using TEM and 3D reconstruction methods (Figure 3) and three different explants of *A. thaliana* for the SE induction showed that the number of PD between cells within the embryogenic symplasmic domain was the highest, followed by that between cells within the meristematic symplasmic domain, but the number of PD on the border of these symplasmic domains was the lowest [88]. During the SE of *Pinus nigra*, PD were only detected between the cells of embryogenic calli [106]. During *D. carota* SE, many PD between embryogenic cells were detected in contrast to cells within non-embryogenic calli, where fewer PD were found [107]. Embryogenic cells of *Cocos nucifera* calli were characterized by a lack of PD in the walls on the border with non-embryogenic cells [108]. In *C. nucifera* callus, more PD between the meristematic cells than between the embryogenic cells were detected [108]. In *Panicum maximum* embryogenic culture, PD were detected only between cells realizing the same developmental program but were absent on the border between cells realizing different programs [109]. The same results were obtained for *Cichorium* spp. [98], *Zea mays* callus [110], and during the early stages of *T. aestivum* pollen embryogenesis [111].

### 3.3. Summary of Symplasmic Isolation during SE and Future Prospects

The contribution of symplasmic isolation to SE regulation seems to be proven. The temporal relationship between callose deposition, followed by expression of the *WOX2* gene, and an association of callose deposition with a decreased *DR5* auxin response in embryogenic cells seems to be a good scenario combining several important elements of the signaling pathway in SE regulation. Despite much research on somatic embryogenesis, there are still many unanswered questions. 

It seems necessary to perform a further search for molecular markers, in the form of characteristic epitopes of cell wall components, in order to understand the role of the cell wall in SE, especially in its early stages when cell proliferation and growth must occur. Obtaining comparable results requires conducting research using transgenic lines in which the appearance of the *WOX2* gene expression can be followed, with a simultaneous analysis of the chemical composition of the wall and/or symplasmic communication. A study of the interaction between the changes in cell wall components and PD seems to be an interesting further direction of research, especially in the sense that we know that changes in pectin presence occur close to PD and affect symplasmic communication [112]. Unfortunately, there is no information about such correlations during SE, where symplasmic isolation plays a pivotal role in the direction of cell differentiation.

A challenge in studying the role of PD in SE regulation is their nano-size and the need for in vivo studies. Only such an experimental approach will allow for the analysis of dynamic changes in the movement of signals between cells. Another challenge is the need to construct fusion proteins (not only GFP, which is not a natural protein for plants) to study the movement of possible TF through PD during SE. This analysis would allow the study of naturally occurring signaling molecules that may be potential regulators of cell differentiation. Another challenge is the microscopic analysis using confocal microscopy, which allows for spatial–temporal studies of the signal movement in vivo. Still, it is sensitive to the thickness of the sample, and the explant is not a single layer of cells, which makes such studies difficult.

Although these aspects of SE have not been discussed in this paper, it is worth mentioning that the research on SE at the molecular level is advanced. Some main signal transduction pathways are known, considering the interconnections between stress, hormones, plant genotype, and culture conditions [113]. However, it seems that attempts to identify key physiological, molecular, and genetic triggers that are valid for all SE systems will be hard to establish [60].

Certainly, putting together the existing data describing the contribution of individual factors regulating SE, such as changes in gene expression and epigenetics, transcription factors, the participation of symplasmic isolation, and chemical changes of cell walls, requires further research before we can obtain a comprehensive picture of the factors involved in SE.

## Figures and Tables

**Figure 1 plants-12-01951-f001:**
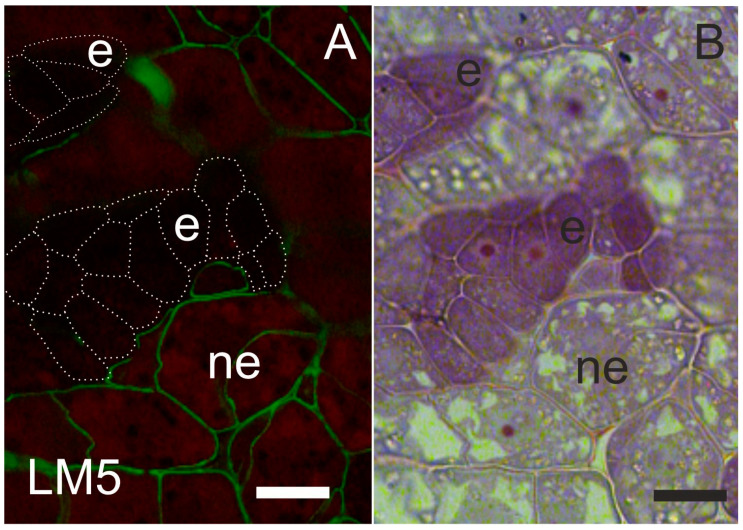
Localization of pectic epitope recognized by LM5 antibody (**A**) in *35S:BBM* Arabidopsis explants and (**B**) the same section after staining with TBO (Toluidine blue O), characterizing the state of development (e—embryogenic cells marked in A by dotted lines, ne—non-embryogenic cells; green color indicates the epitope presence in cell walls; red fluorescence indicates chlorophyll; scale bar: 20 µm).

**Figure 2 plants-12-01951-f002:**
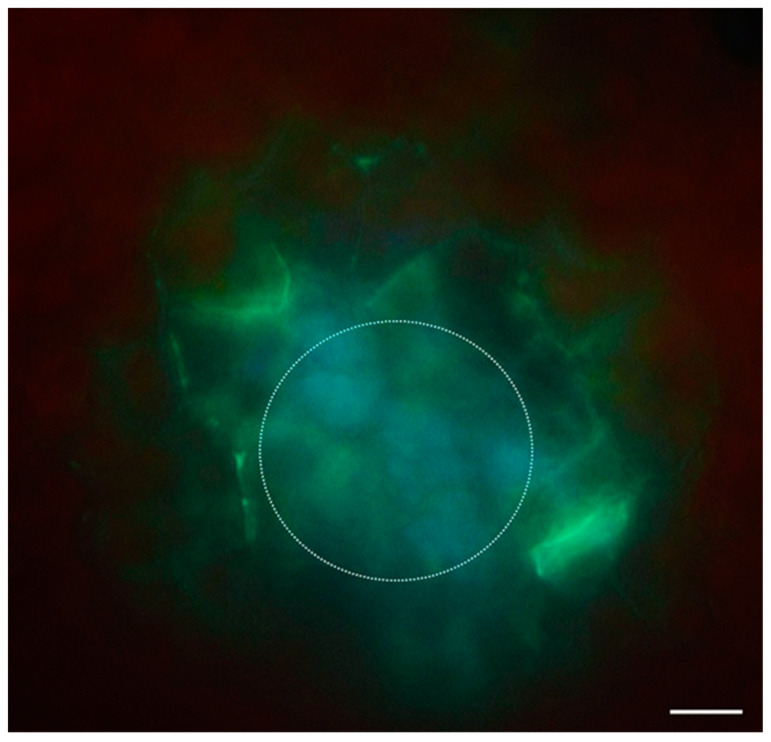
The callose localization in explant cells of *35S:BBM* Arabidopsis during SE. Callose (green color) is deposited in the walls of embryogenic cells (white circle) located at the border with the non-embryogenic cells (scale bar—20 µm).

**Figure 3 plants-12-01951-f003:**
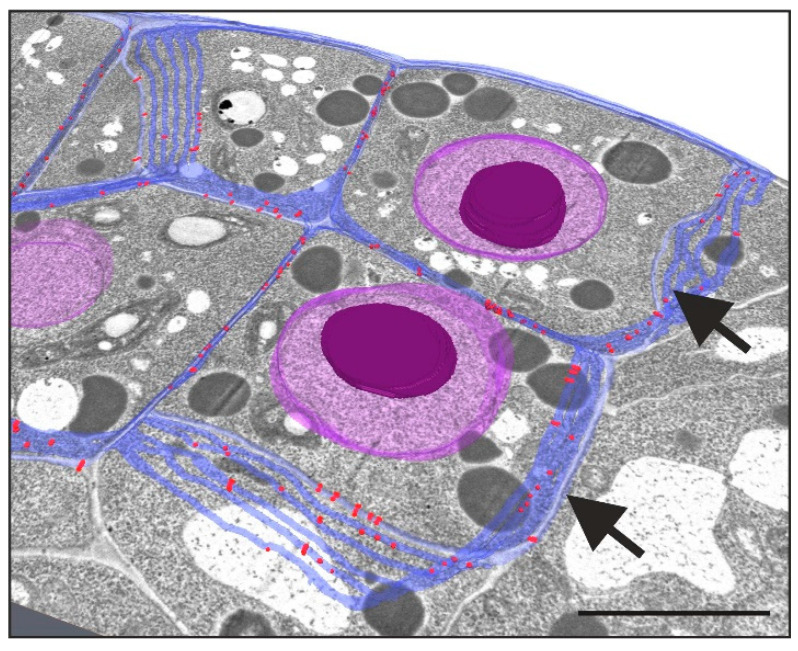
3D visualization of PD distribution in walls of embryogenic cells (arrows) of Arabidopsis explant (violet—nucleus with nucleolus; blue—cell walls; red dots—PD; scale bar—4 µm).

## Data Availability

Data are contained within the article.

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
