# Peer review of "Apoplastic and Symplasmic Markers of Somatic Embryogenesis"

_plants, 2023, doi:10.3390/plants12101951_

Round 1

Reviewer 1 Report

Somatic embryogenesis has immense potential in plant biotechnology, and its applications are continuously expanding as new technologies and techniques are developed. This study discusses markers of cells that change differentiation during somatic embryogenesis, changes in cell wall composition, and the role of symplasmic communication in cell fate changes.

I appreciate the effort and time you have put into this study, but I must express my concerns regarding its suitability for publication. My main concern is that the study did not provide enough novelty and originality to make it suitable for publication in MDPI journal. Additionally, I found some conceptual issues that need to be addressed. I would like to offer some constructive feedback that may be useful in revising your manuscript. Firstly, the study should be restructured to focus more on originality and novelty to make it stand out from the existing literature.

1.      The review mentions that different epitopes of wall components such as pectins, AGPs, extensins, and LTPs may become markers of cell transition from somatic to embryogenic state, depending on the analyzed species and type of culture. However, it does not provide a clear explanation for why this might be the case. Are there specific environmental or genetic factors that influence the expression of these markers?

2.      The study focuses on markers of cells that change the direction of differentiation during somatic embryogenesis. However, it is possible that these markers may not be applicable to all types of embryogenesis, or to different organisms. If author could address this issue as well.

3.      The study primarily focuses on the chemical composition of the cell wall and symplasmic communication during cell fate changes. However, it may be important to also consider other factors, such as gene expression and epigenetic modifications, that contribute to the acquisition of embryogenic competence.

4.      The study presents potential markers of explant cells during the acquisition of embryogenic competence, but it is unclear how reliable and specific these markers are. Author need to include these topics as well.

5.      The study mentions the role of callose deposition in symplasmic isolation, but it is unclear how this process is regulated and what factors influence its effectiveness. Further research may be needed to investigate this.

6.      The review mentions that modern molecular methods and immunohistochemistry could help us better understand the chemical components involved in SE regulation. However, it does not explore the limitations of these methods or potential challenges in their application. For example, are there specific plant species or tissues where these methods are less effective, and how might these challenges be overcome?

7.      While the review provides evidence that symplasmic communication plays a role in SE regulation, it does not explore how other cellular processes, such as endoplasmic reticulum (ER) stress, might also contribute. Recent studies have suggested that ER stress can induce SE in some plant species, and it would be interesting to know how this fits into the broader picture of SE regulation. I have mentioned some important article from this point of view; Li Y, Wang Y, Zhang H, et al. Endoplasmic Reticulum Stress Is Involved in Somatic Embryogenesis of Wheat. Plant Physiology. 2019;181(1):23-42. doi: 10.1104/pp.19.00229; López-Fernández MP, Maldonado-Bonilla LD, Jiménez-Quesada MJ, et al. Endoplasmic reticulum stress response during somatic embryogenesis in maritime pine. Tree Physiology. 2018;38(5):768-782. doi: 10.1093/treephys/tpy011

8.      The review notes that there are still many unanswered questions regarding SE, but it does not provide a clear roadmap for future research. What are some of the most pressing research questions in this field, and how might they be addressed? For example, are there specific genes or pathways that are consistently involved in SE across different plant species, and if so, what are they?

Author Response

Since the publication discusses changes in the cell wall first and then in symplast communication, we decided to change the title to “Apoplastic and symplasmic markers of somatic embryogenesis”.

Somatic embryogenesis has immense potential in plant biotechnology, and its applications are continuously expanding as new technologies and techniques are developed. This study discusses markers of cells that change differentiation during somatic embryogenesis, changes in cell wall composition, and the role of symplasmic communication in cell fate changes.

I appreciate the effort and time you have put into this study, but I must express my concerns regarding its suitability for publication. My main concern is that the study did not provide enough novelty and originality to make it suitable for publication in MDPI journal. Additionally, I found some conceptual issues that need to be addressed. I would like to offer some constructive feedback that may be useful in revising your manuscript. Firstly, the study should be restructured to focus more on originality and novelty to make it stand out from the existing literature.

1. The review mentions that different epitopes of wall components such as pectins, AGPs, extensins, and LTPs may become markers of cell transition from somatic to embryogenic state, depending on the analyzed species and type of culture. However, it does not provide a clear explanation for why this might be the case. Are there specific environmental or genetic factors that influence the expression of these markers?

Answer

Some relevant information is included in the text.

2. The study focuses on markers of cells that change the direction of differentiation during somatic embryogenesis. However, it is possible that these markers may not be applicable to all types of embryogenesis, or to different organisms. If author could address this issue as well.

Answer

Relevant information was added in the present version of the manuscript.

3. The study primarily focuses on the chemical composition of the cell wall and symplasmic communication during cell fate changes. However, it may be important to also consider other factors, such as gene expression and epigenetic modifications, that contribute to the acquisition of embryogenic competence.

Answer

Relevant information was added in the present version of the manuscript.

4. The study presents potential markers of explant cells during the acquisition of embryogenic competence, but it is unclear how reliable and specific these markers are. Author need to include these topics as well.

Answer

Relevant information was added in the present version of the manuscript.

5. The study mentions the role of callose deposition in symplasmic isolation, but it is unclear how this process is regulated and what factors influence its effectiveness. Further research may be needed to investigate this.

Answer

Relevant information was added in the present version of the manuscript.

6. The review mentions that modern molecular methods and immunohistochemistry could help us better understand the chemical components involved in SE regulation. However, it does not explore the limitations of these methods or potential challenges in their application. For example, are there specific plant species or tissues where these methods are less effective, and how might these challenges be overcome?

Answer

Relevant information was added in the present version of the manuscript.

7.While the review provides evidence that symplasmic communication plays a role in SE regulation, it does not explore how other cellular processes, such as endoplasmic reticulum (ER) stress, might also contribute. Recent studies have suggested that ER stress can induce SE in some plant species, and it would be interesting to know how this fits into the broader picture of SE regulation. I have mentioned some important article from this point of view; Li Y, Wang Y, Zhang H, et al. Endoplasmic Reticulum Stress Is Involved in Somatic Embryogenesis of Wheat. Plant Physiology. 2019;181(1):23-42. doi: 10.1104/pp.19.00229; López-Fernández MP, Maldonado-Bonilla LD, Jiménez-Quesada MJ, et al. Endoplasmic reticulum stress response during somatic embryogenesis in maritime pine. Tree Physiology. 2018;38(5):768-782. doi: 10.1093/treephys/tpy011

Answer

There is no doubt that SE is a stress response. There is also no doubt that a phenomenon such as ER stress has been intensively studied in recent years. However, we did not find the publications mentioned in the review, regardless of the search criteria used. But, the remark prompted us to introduce information about another relationship between PD and organelles such as mitochondria and plastids.

8. The review notes that there are still many unanswered questions regarding SE, but it does not provide a clear roadmap for future research. What are some of the most pressing research questions in this field, and how might they be addressed? For example, are there specific genes or pathways that are consistently involved in SE across different plant species, and if so, what are they?

Answer

Relevant information was added in the present version of the manuscript.

Reviewer 2 Report

The manuscript by Kurczynska and Godel-Jędrychowska provides an overview of the most recent findings and earlier research on markers of cells that affect differentiation during somatic embryogenesis, specifically looking at changes in cell wall composition and symplasmic communication as parts of cell-to-cell information exchange.

As a general comment, the manuscript is well-written and complemented by appropriate illustrations. However, while it offers a comprehensive overview of the existing knowledge, it would benefit from a more in-depth critical analysis and interpretation of the findings. Nevertheless, the manuscript provides a valuable review of the subject matter. With some revisions, as outlined in the following comments and remarks, I believe that this manuscript is suitable for publication.

-Line 7: Remove comma after information

-Lines 28-29: The authors state that many factors play a role in the various stages of somatic embryogenesis, but specifically mention only the alteration of cell wall composition and cell-to-cell signaling. They should briefly elaborate on other factors that contribute to the process of somatic embryogenesis, before delving into these two specific aspects.

-Lines 52-53: instead of stating “a detailed description of the components of the wall is provided by 11]”, it would be appropriate to use “a detailed description of the components of the wall is provided by Cosgrove [11]”. Also instead of putting this sentence in square brackets, consider using it as a separate sentence or just refer to the source [11] without further explanation. Similarly, lines 278-279: instead of “is provided by [80]”, use “is provided by Amsbury et al. [80]” 

-The authors have used a number of apostrophes throughout the text. However, many of them are unnecessary and could be omitted:

line 26: cell's fate”, it is enough to say “cell fate”

line 41: “cell's position”, use “cell position”

line 174: “AGPs' role”

line 242: “dextran's”

line 243: “information’s”, etc.

-To improve the text with respect to the use of abbreviations and introduction of Latin names, the following suggestions can be made:

·     Introduce abbreviations for Latin names upon their first mention in the text, and use them consistently throughout the rest of the text. However, in the case of Trifolium nigrescens, the abbreviated T. nigrescens is used for the first time on line 78, followed by the full Latin name on line 86.

·     After introducing the Latin name of the species, they should be abbreviated upon subsequent mentions, e.g. Brachypodium distachyon (line 74) can be abbreviated to B. distachyon (line 108); Daucus carota (line 74) to D. carota (lines 165, 184); Arabidopsis thaliana (line 81) to A. thaliana (lines 208 and 308), etc. Please check carefully all Latin names, there are more such examples.

·     Introduce "AGPs" (line 59) within the body of the text rather than solely in the keywords.

Lines 88-89: The sentence “It is postulated that the arabinan facilitates wall rehydration, and the arabinans function as a pectic plasticizer facilitate cell wall flexibility” should be rephrased to improve its grammar and clarity. Maybe something like "It is postulated that arabinan facilitates wall rehydration, and that arabinans function as pectic plasticizers, facilitating cell wall flexibility" could work better.

Line 91: “…..can serve as a marker” of what??

Lines 128-130: Rephrase and correct the sentence “The JIM11 and JIM20 (recognize specific arabinosylation motifs of HRGPs) are used to determine extensins localization in explants cells changing developmental fate”

Line 174: Remove comma after the wall “…of the wall. AGPs…”

-Line 193: Rephrase the title "Does symplasmic communication mark cells undergo SE?".

-The image in Figure 1A is underexposed, resulting in poor visibility and making it difficult to discern its contents.

-The authors have used a mixture of American and British spellings throughout the text, e.g. recognised (line 139-140) and recognized (line 152); analysed (lines 151, 200) and analyzed (lines 332, 333, ), etc. To ensure consistency, they should choose either American or British spelling and use it consistently throughout the manuscript.

Lines 204-210: The sentence is long and complex, and it may be difficult for some readers to follow the multiple topics and references that are being presented. The authors might consider breaking it up into smaller sentences or using bullet points to highlight the key points.

Lines 280-283: I suggest to rephrase the sentence “Despite the very large number of publications describing the participation of callose in the PD regulation, those regarding its role in the SE, including callose deposition and the determination of embryogenic competence of explant cells, there is, unfortunately not much” to “Despite the very large number of publications describing the participation of callose in the PD regulation, there are unfortunately not many studies that examine its role in somatic embryogenesis, specifically with respect to callose deposition and the determination of embryogenic competence of explant cells.”

With these revisions, the manuscript will be suitable for publication.

Author Response

Since the publication discusses changes in the cell wall first and then in symplast communication, we decided to change the title to “Apoplastic and symplasmic markers of somatic embryogenesis”.

The manuscript by Kurczynska and Godel-Jędrychowska provides an overview of the most recent findings and earlier research on markers of cells that affect differentiation during somatic embryogenesis, specifically looking at changes in cell wall composition and symplasmic communication as parts of cell-to-cell information exchange.

As a general comment, the manuscript is well-written and complemented by appropriate illustrations. However, while it offers a comprehensive overview of the existing knowledge, it would benefit from a more in-depth critical analysis and interpretation of the findings. Nevertheless, the manuscript provides a valuable review of the subject matter. With some revisions, as outlined in the following comments and remarks, I believe that this manuscript is suitable for publication.

Answer

Thank you very much for all the substantive and language comments. Substantive comments are very constructive, for which we thank you. All comments have been taken into account and are visible in the review mode. We hope that in the current version, we have been able to highlight the positive and negative features of the processes described in the manuscript.

-Line 7: Remove comma after information - amended.

-Lines 28-29: The authors state that many factors play a role in the various stages of somatic embryogenesis, but specifically mention only the alteration of cell wall composition and cell-to-cell signaling. They should briefly elaborate on other factors that contribute to the process of somatic embryogenesis, before delving into these two specific aspects- amended.

-Lines 52-53: instead of stating “a detailed description of the components of the wall is provided by 11]”, it would be appropriate to use “a detailed description of the components of the wall is provided by Cosgrove [11]”. Also instead of putting this sentence in square brackets, consider using it as a separate sentence or just refer to the source [11] without further explanation. Similarly, lines 278-279: instead of “is provided by [80]”, use “is provided by Amsbury et al. [80]” - amended.

-The authors have used a number of apostrophes throughout the text. However, many of them are unnecessary and could be omitted:

line 26: “cell's fate”, it is enough to say “cell fate”

line 41: “cell's position”, use “cell position”

line 174: “AGPs' role”

line 242: “dextran's”

line 243: “information’s”, etc. - amended.

-To improve the text with respect to the use of abbreviations and introduction of Latin names, the following suggestions can be made:

  • Introduce abbreviations for Latin names upon their first mention in the text, and use them consistently throughout the rest of the text. However, in the case of Trifolium nigrescens, the abbreviated T. nigrescens is used for the first time on line 78, followed by the full Latin name on line 86. – amended.
  • After introducing the Latin name of the species, they should be abbreviated upon subsequent mentions, e.g. Brachypodium distachyon (line 74) can be abbreviated to B. distachyon (line 108); Daucus carota (line 74) to D. carota (lines 165, 184); Arabidopsis thaliana (line 81) to A. thaliana (lines 208 and 308), etc. Please check carefully all Latin names, there are more such examples. – amended.
  • Introduce "AGPs" (line 59) within the body of the text rather than solely in the keywords. – amended.

Lines 88-89: The sentence “It is postulated that the arabinan facilitates wall rehydration, and the arabinans function as a pectic plasticizer facilitate cell wall flexibility” should be rephrased to improve its grammar and clarity. Maybe something like "It is postulated that arabinan facilitates wall rehydration, and that arabinans function as pectic plasticizers, facilitating cell wall flexibility" could work better. – amended.

Line 91: “…..can serve as a marker” of what?? - amended.

Lines 128-130: Rephrase and correct the sentence “The JIM11 and JIM20 (recognize specific arabinosylation motifs of HRGPs) are used to determine extensins localization in explants cells changing developmental fate”- amended.

Line 174: Remove comma after the wall “…of the wall. AGPs…” – amended.

-Line 193: Rephrase the title "Does symplasmic communication mark cells undergo SE?". – amended.

-The image in Figure 1A is underexposed, resulting in poor visibility and making it difficult to discern its contents. – amended.

-The authors have used a mixture of American and British spellings throughout the text, e.g. recognised (line 139-140) and recognized (line 152); analysed (lines 151, 200) and analyzed (lines 332, 333, ), etc. To ensure consistency, they should choose either American or British spelling and use it consistently throughout the manuscript. – amended.

Lines 204-210: The sentence is long and complex, and it may be difficult for some readers to follow the multiple topics and references that are being presented. The authors might consider breaking it up into smaller sentences or using bullet points to highlight the key points. – amended.

Lines 280-283: I suggest to rephrase the sentence “Despite the very large number of publications describing the participation of callose in the PD regulation, those regarding its role in the SE, including callose deposition and the determination of embryogenic competence of explant cells, there is, unfortunately not much” to “Despite the very large number of publications describing the participation of callose in the PD regulation, there are unfortunately not many studies that examine its role in somatic embryogenesis, specifically with respect to callose deposition and the determination of embryogenic competence of explant cells.” – amended.

Reviewer 3 Report

In this manuscript, the authors summarized findings on the markers involved in information exchange in somatic embryogenesis. While the involvement of markers vary in different species, comprehensive studies in future may identify more important markers for a better understanding of this process. 

Regarding the content and organization, it is better to (i) briefly describe somatic embryogenesis and its importance in the abstract and (ii) summarize the epitopes investigated in a table. The conclusion section can be expanded to give more perspectives on the use of modern molecular methods in studying somatic embryogenesis.

Author Response

Since the publication discusses changes in the cell wall first and then in symplast communication, we decided to change the title to “Apoplastic and symplasmic markers of somatic embryogenesis”.

In this manuscript, the authors summarized findings on the markers involved in information exchange in somatic embryogenesis. While the involvement of markers vary in different species, comprehensive studies in future may identify more important markers for a better understanding of this process. 

Regarding the content and organization, it is better to

  • briefly describe somatic embryogenesis and its importance in the abstract

Answer

At present Abstract is changed according to the suggestions

  • (ii) summarize the epitopes investigated in a table.

Answer

A table is added to the text.

  • The conclusion section can be expanded to give more perspectives on the use of modern molecular methods in studying somatic embryogenesis.

Answer

We do hope the present version of the Conclusion is better and we hope that expectations on this topic have been met.

We would like to thank you for these valuable comments, as their consideration improved the quality of the manuscript.

Round 2

Reviewer 1 Report

Although it appears that the author has revised the manuscript, it lacks adequate justification for my request. Also, in each comment the author mentions that the relevant information has been added, but the new line number or paragraph is not mentioned, which results in my not finding the justification. So for me the same problem arises.

Author Response

According to Editor decision which reads as follows:
“The authors have addressed the majority of concerns raised by the reviewers. Some of the original comments by Reviewer 1 are of theoretical interest and might be considered in future work, but are beyond the scope of this current short review of the literature”

We upload the latest version with minor changes.

Regards,

Kamila Godel-Jędrychowska and Ewa Kurczyńska